# LDSA: Learning Dynamic Subtask Assignment in Cooperative Multi-Agent Reinforcement Learning

**Mingyu Yang**[1]**, Jian Zhao**[1]**, Xunhan Hu**[1]**, Wengang Zhou**[1,3][†]**, Jiangcheng Zhu**[2]**, Houqiang Li**[1,3][†]

[1]University of Science and Technology of China, [2]Huawei Cloud
[3]Hefei Comprehensive National Science Center, Institute of Artificial Intelligence
{ymy,zj140,cathyhxh}@mail.ustc.edu.cn
zhujiangcheng@huawei.com, {zhwg,lihq}@ustc.edu.cn

## Abstract

Cooperative multi-agent reinforcement learning (MARL) has made prominent progress in recent years. For training efficiency and scalability, most of the MARL algorithms make all agents share the same policy or value network. However, in many complex multi-agent tasks, different agents are expected to possess specific abilities to handle different subtasks. In those scenarios, sharing parameters indiscriminately may lead to similar behavior across all agents, which will limit the exploration efficiency and degrade the final performance. To balance the training complexity and the diversity of agent behavior, we propose a novel framework to learn dynamic subtask assignment (LDSA) in cooperative MARL. Specifically, we first introduce a subtask encoder to construct a vector representation for each subtask according to its identity. To reasonably assign agents to different subtasks, we propose an ability-based subtask selection strategy, which can dynamically group agents with similar abilities into the same subtask. In this way, agents dealing with the same subtask share their learning of specific abilities and different subtasks correspond to different specific abilities. We further introduce two regularizers to increase the representation difference between subtasks and stabilize the training by discouraging agents from frequently changing subtasks, respectively. Empirical results show that LDSA learns reasonable and effective subtask assignment for better collaboration and significantly improves the learning performance on the challenging StarCraft II micromanagement benchmark and Google Research Football.

## 1 Introduction

Cooperative multi-agent reinforcement learning (MARL) has recently received much attention due to its broad prospects on many real-world challenging problems, such as traffic light control [1], autonomous cars [2] and robot swarm control [3]. Compared to single-agent scenarios, multi-agent tasks pose more challenges. On the one hand, the observation transition function of each agent is related to the policies of other agents, which are constantly updated during training. Hence, from the perspective of any individual agent, the environment is extremely non-stationary, which will be exacerbated with more agents. On the other hand, the joint action-observation space of the multi-agent task grows exponentially with the number of agents. These two issues prevent MARL algorithms from scaling to more agents.

To cope with a large number of agents, most of the recent MARL works, including value-based [4–9] and policy gradient [10–15], utilize a technique of *policy decentralization with shared parameters* [16],

---

[†]Corresponding authors: Wengang Zhou and Houqiang Li

36th Conference on Neural Information Processing Systems (NeurIPS 2022).

whereby all agents share a decentralized policy or value network. There are several merits that make parameter sharing so popular in MARL. First and foremost, it considerably reduces the total number of trainable parameters and makes the learning complexity tractable. Besides, agents with the same parameters can share training experience with each other, which can effectively mitigate the slowdown of convergence rate due to non-stationarity [17]. However, many complex multi-agent tasks consist of a set of subtasks, where the transition or reward functions are distinct [18, 19]. Solving every subtask requires some specific abilities. In contrast, fully-shared parameters may cause all agents to behave similarly and hinder the diversity of agents' policies. For example, on the Google Research Football task [20], all agents will compete for the ball if sharing parameters [16]. The similar behaviors across all agents will limit the exploration efficiency and degrade the final performance [21]. Therefore, it's difficult for a single neural network to learn the specific abilities required by all subtasks.

Alternatively, another solution is to learn a separate policy for each agent without any parameter sharing, which allows diverse policies but leads to high training complexity. To balance the training complexity and the diversity of agents' behaviors, a better method is to learn a distinct neural network for each subtask and group the agents by subtasks. Agents dealing with the same subtask typically learn similar policies and thus can share their training experiences to accelerate training. However, this method poses two new problems: (1) how to decompose a multi-agent task into subtasks and (2) how to assign agents to subtasks. Most of the previous works [22–25] predefine the task decomposition by using a rich prior knowledge of application domains, which may not be available in practice. To the best of our knowledge, only one recent work, RODE [26], learns explicit task decomposition without prior domain knowledge. RODE defines the subtasks based on joint action space decomposition during pretraining. Specifically, RODE learns effect-based action representations to cluster actions and then treats each cluster of actions as a subtask, where each subtask only needs to fulfill the functionality of a subset of actions. But it may fail to solve subtasks when some basic actions are necessary for all subtasks, such as the actions of moving in the StarCraft II micromanagement tasks [27]. What's more, the effect of each action changes dynamically with the environment and thus it may be difficult to determine the actions' effect only through pretraining.

In this work, we propose a novel framework to learn dynamic subtask assignment (LDSA) in cooperative MARL. Our method first proposes a subtask encoder that constructs a vector representation for each subtask according to its identity. The action-observation history typically reflects the behavioral habits and potential abilities of each agent, which is an important clue for selecting subtasks. Therefore, we employ a trajectory encoding network to obtain the action-observation history of each agent. Then, for every timestep, each agent acquires a categorical distribution of subtask selection based on the cosine similarity of its action-observation history and representations of all subtasks, and samples a subtask using Gumbel-Softmax [28] for training, which is a reparameterization trick allowing backpropagate through samples. After that, we learn a separate policy for each subtask and enable agents dealing with the same subtask to share their learning. To associate the policy of each subtask with its representation, we introduce a subtask decoder to generate the policy parameters of each subtask based on its representation, which can also avoid similar policies between different subtasks. Furthermore, we introduce two regularizers to increase the representation difference between subtasks and avoid agents changing subtasks frequently to stabilize training, respectively.

We evaluate our method on the challenging StarCraft II micromanagement tasks (SMAC) [27]. The results show that our LDSA significantly improves the learning performance on the SMAC benchmark compared to the baselines, especially on *Hard* and *Super Hard* scenarios. Additional experiments on Google Research Football (GRF) [20] further demonstrate the effectiveness of LDSA on various multi-agent tasks. The ablation studies confirm the benefits of the two regularizers and the high efficiency of LDSA. Moreover, visualizations on SMAC reveal that LDSA could reasonably and effectively decomposes the task with dynamic subtask assignment for better collaboration.

## 2 Preliminaries

In this section, we introduce the necessary background knowledge to understand this paper. We first describe the problem formulation and the definition of subtasks for a fully cooperative multi-agent task. Then, we present the value function factorization methods with the centralized training with decentralized execution (CTDE) paradigm [29, 30], which will be adopted in this paper.

## 2.1 Problem formulation

This work is focused on a fully cooperative multi-agent task with only partial observation for each agent. This task is typically modeled as a decentralized partially observable markov decision process (Dec-POMDP) [31] defined by a tuple $G = \langle A, S, U, P, r, Z, O, \gamma \rangle$, where $A \equiv \{g_1, g_2, \cdots, g_n\}$ denotes the finite set of agents, $\gamma \in [0, 1)$ is the discount factor and $S$ is the set of global state of the environment. At each timestep, each agent $g_a \in A$ selects an action $u_a \in U$, forming a joint action $\mathbf{u} \in U^n$, where $a \in \{1, 2, \cdots, n\}$ is the agent identity. This leads to a transition on the environment according to the state transition function $P(s'|s, \mathbf{u}) : S \times U^n \times S \to [0, 1]$ and all agents receive a shared team reward $r(s, \mathbf{u}) : S \times U^n \to \mathbb{R}$, where $s \in S$ describes the global state. Due to the partial observability, each agent $g_a$ only obtains a partial observation $o_a \in Z$ according to the observation function $O(s, g_a) : S \times A \to Z$ and holds an action-observation history $\tau_a \in T \equiv (Z \times U)^*$. The joint action-observation history of all agents is denoted as $\boldsymbol{\tau}$.

To achieve scalability, a popular approach adopted by many recent MARL works [4–8] is that each agent $g_a$ constructs a decentralized policy $\pi_a(u_a|\tau_a; \theta)$ with shared parameters $\theta$. However, such simple parameter sharing may fail to deal with many complex multi-agent tasks that require diverse abilities among agents. To this end, we propose to decompose a fully cooperative multi-agent task into subtasks. We present the definition of subtasks in the following.

**Definition 1** (Subtasks). *For a fully cooperative multi-agent task $G = \langle A, S, U, P, r, Z, O, \gamma \rangle$, we assume there exists a set of $k$ subtasks, denoted as $\Phi \equiv \{\phi_1, \phi_2, \cdots, \phi_k\}$, where $k \in \mathbb{N}^+$ is unknown and we consider $k$ as a tunable hyperparameter. Each subtask $\phi_i$ is defined by a tuple $\langle x_{\phi_i}, G_{\phi_i}, \pi_{\phi_i} \rangle$, where $i \in \{1, 2, \cdots, k\}$ is the identity of subtask, $x_{\phi_i} \in \mathbb{R}^m$ is a vector representation (or latent embeddings) of subtask $\phi_i$, $G_{\phi_i} = \langle A_{\phi_i}, S, U, P, r, Z, O, \gamma \rangle$ is the Dec-POMDP model of subtask $\phi_i$, and $\pi_{\phi_i} : T \times U \to [0, 1]$ is the policy of subtask $\phi_i$. $A_{\phi_i}$ is the set of agents assigned to subtask $\phi_i$ and each agent can only select one subtask to solve at each timestep, i.e., $\bigcup_{i=1}^k A_{\phi_i} = A$ and $A_{\phi_i} \bigcap A_{\phi_j} = \varnothing$ if $i \neq j$. Each agent $g_a \in A_{\phi_i}$ shares the policy parameters of $\pi_{\phi_i}$.*

Our objective is to learn a optimal set of subtasks $\Phi^*$ so as to maximize the expected global return $Q_{tot}^\Phi(\boldsymbol{\tau}, \mathbf{u}) = \mathbb{E}_{s_{1:\infty}, \mathbf{u}_{1:\infty}} [\sum_{t=0}^\infty \gamma^t r_t | s_0 = s, \mathbf{u}_0 = \mathbf{u}, \Phi]$, where $r_t$ is the team reward at timestep $t$. Therefore, we are required to learn the vector representation $x_{\phi_i}$, the agent assignment $A_{\phi_i}$ and the policy $\pi_{\phi_i}$ for each subtask $\phi_i$, which will be introduced in detail in Sec. 3.

## 2.2 Value function factorization with CTDE paradigm

To make policy learning stable and scalable in MARL, the paradigm of CTDE [29, 30] has been proposed and gained substantial attention. In this paradigm, agents learn their policies together with access to global information during training in a centralized way and only rely on their local observations during execution in a decentralized way. The CTDE paradigm has been exploited by many recent MARL algorithms [4–8, 10, 11], among which value function factorization methods [4–8] have shown superior performance on challenging cooperative multi-agent tasks [27]. These methods factorize the global Q-value function $Q_{tot}$ into individual Q-value functions $Q_a$ for each agent $g_a$, where $Q_a$ is only based on the local action-observation history $\tau_a$ for decentralized execution. To guarantee the consistency between greedy action selection in global and individual Q-values, such factorization has to satisfy Individual-Global-Maximum (IGM) principle [6] as follows:

$$\arg \max_{\mathbf{u}} Q_{tot}(\boldsymbol{\tau}, \mathbf{u}) = \left( \arg \max_{u_1} Q_1(\tau_1, u_1), \cdots, \arg \max_{u_n} Q_n(\tau_n, u_n) \right). \tag{1}$$

In this work, we consider a representative value function factorization method, QMIX [5], which estimates global Q-value as a non-linear combination of individual Q-values as follows:

$$Q_{tot}(\boldsymbol{\tau}, \mathbf{u}) = f_w \left( Q_1(\tau_1, u_1), \cdots, Q_n(\tau_n, u_n) | s \right), \tag{2}$$

where $f_w$ is a monotonic function that conditions on the global state $s$, i.e., $\frac{\partial f_w}{\partial Q_a} \geq 0, \forall a \in \{1, 2, \cdots, n\}$. Although Eq. 2 is only sufficient and unnecessary to IGM, QMIX is a lightweight and efficient method showing state-of-the-art performance on SMAC [27].

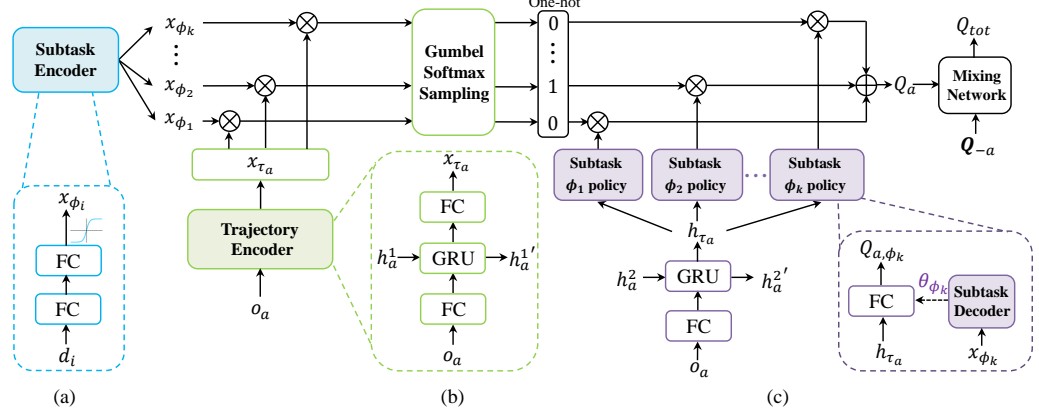

Figure 1: Overview of LDSA framework. (a) The subtask representation learning structure (shown in blue). (b) Architecture of subtask selection for each agent (shown in green). (c) The policy learning for each subtask (shown in purple).

## 3  Method

In this section, we present the LDSA learning framework as shown in Fig. 1. We first introduce how to construct a set of distinct subtasks for decomposing a multi-agent task. Next, we discuss how each agent selects a subtask according to its abilities. After grouping agents by subtasks, we present the policy of each subtask that is representation-dependent. Finally, we show the overall training objective and the inference strategy.

### 3.1  Distinct subtask representation

Many complex multi-agent tasks involve a set of subtasks that have different responsibilities. For example, running a company requires multiple departments to work together. Previous work typically leverages the prior domain knowledge to decompose a complex task, which is not practical for many uncertain environments. To remove the dependence on prior knowledge and apply it to broader multi-agent tasks, we propose to learn a vector representation $x_{\phi_i}$ for each subtask $\phi_i$ according to its identity $i$.

Specifically, we use a two-layer fully-connected (FC) network with a tanh activation function in Fig. 1(a) to learn a subtask encoder $f_e(\cdot; \theta_e) : \mathbb{R}^k \to \mathbb{R}^m$, parameterized by $\theta_e$. The subtask encoder maps the one-hot identity $d_i \in \mathbb{R}^k$ of subtask $\phi_i$ to an $m$-dimensional representation space. The tanh activation function is to constraint the value range of the representation space. Then, we employ the subtask encoder to construct a vector representation $x_{\phi_i} \in \mathbb{R}^m$ for each subtask $\phi_i$, *i.e.*,

$$x_{\phi_i} = f_e(d_i; \theta_e), \forall i \in \{1, 2, \cdots, k\}. \tag{3}$$

If subtasks are similar, the task decomposition makes little sense. Therefore, to keep differences between subtasks, we propose a regularizer that maximizes the $L_2$ distance between representations of different subtasks as follows:

$$\mathcal{L}_\phi(\theta_e) = \mathbb{E}_\mathcal{D}\Big[ - \sum_{i \neq j} \|x_{\phi_i} - x_{\phi_j}\|^2 \Big], \tag{4}$$

where $\mathcal{D}$ is the replay buffer. The subtask representation learning continues throughout the training process, which can automatically adapt to the dynamic changes in the environment. The overall optimization objective of the subtask encoder will be introduced in Sec. 3.4.

### 3.2  Ability-based subtask selection

With the representation of subtasks, we need to design a subtask selection strategy for each agent based on its ability. In real life, we can easily infer a person's role from the trajectory of his or her

behavior. Similarly, the action-observation history of an agent can reflect its behavioral habits and potential abilities. Therefore, we utilize a shared trajectory encoder (shown in Fig. 1(b)) consisting of a GRU [32] and two fully-connected networks to obtain the action-observation history of each agent. The trajectory encoder $f_h(\cdot; \theta_h)$, parameterized by $\theta_h$, encodes the action-observation history of each agent $g_a$ into a vector $x_{\tau_a} \in \mathbb{R}^m$, which has the same length with the subtask representation. Then, agent $g_a$ treats $x_{\tau_a}$ as its ability representation and selects a subtask to solve based on $x_{\tau_a}$.

In our implementation, for each agent $g_a$, we first calculate the cosine similarity of its action-observation history representation $x_{\tau_a}$ and representations of all subtasks $x_\Phi := [x_{\phi_i}]_{i=1}^k$, *i.e.*, $similarity(x_{\tau_a}, x_{\phi_i}) = (x_{\tau_a}^{\mathrm{T}} x_{\phi_i})/(\|x_{\tau_a}\|\|x_{\phi_i}\|)$. Since representations of all subtasks have the same value range through the tanh activation function, we use $x_{\tau_a}^{\mathrm{T}} x_{\phi_i}$ to approximate the cosine similarity for simplicity. Then, we employ softmax function on the cosine similarity and obtain a categorical distribution of subtask selection $\boldsymbol{p}(\Phi|x_{\tau_a}, x_\Phi) := [p(\phi_i|x_{\tau_a}, x_\Phi)]_{i=1}^k$. $p(\phi_i|x_{\tau_a}, x_\Phi)$ is the probability that agent $g_a$ selects subtask $\phi_i$, given by:

$$p(\phi_i|x_{\tau_a}, x_\Phi) = \frac{exp(x_{\tau_a}^{\mathrm{T}} x_{\phi_i})}{\sum_{j=1}^k exp(x_{\tau_a}^{\mathrm{T}} x_{\phi_j})}, \forall i \in \{1, 2, \cdots, k\}, \tag{5}$$

where $exp(\cdot)$ is the exponential function. Sampling a subtask directly from the categorical distribution $\boldsymbol{p}(\Phi|x_{\tau_a}, x_\Phi)$ is not differentiable. To make the subtask selection process trainable, we use the Straight-Through Gumbel-Softmax Estimator [28] to sample a subtask $\phi_j$ that will be discretized as a $k$-dimensional one-hot vector, *i.e.*, the one-hot subtask identity $d_j$.

For every timestep, each agent will select a subtask to dynamically cooperate with each other. This dynamic subtask assignment scheme may make training unstable when an agent frequently changes the subtask in an episode. To smooth the agents' subtask selection and stabilize training, we introduce a second regularizer to minimize the KL divergence between the subtask selection distributions for any two adjacent timesteps as follows:

$$\mathcal{L}_h(\theta_e, \theta_h) = \mathbb{E}_\mathcal{D} \left[ \sum_a D_{\mathrm{KL}} \left( \boldsymbol{p}(\Phi|x_{\tau_a}, x_\Phi) \| \boldsymbol{p}'(\Phi'|x_{\tau_a}', x_\Phi') \right) \right], \tag{6}$$

where $\boldsymbol{p}'(\Phi'|x_{\tau_a}', x_\Phi')$ is the subtask selection distribution at the next timestep, $D_{\mathrm{KL}}(\cdot\|\cdot)$ is the KL divergence operator and the sum is performed across all agents.

### 3.3 Representation-dependent subtask policy

After grouping agents to different subtasks based on their abilities, we learn the policy for each subtask which is illustrated in Fig. 1(c). Generally, agents dealing with the same subtask share the policy parameters and different subtasks have distinct policy parameters. To this end, we first use a new shared trajectory encoder $f_\tau(\cdot|\theta_\tau)$ with parameters $\theta_\tau$ to generate the action-observation history of each agent $g_a$, denoted as $h_{\tau_a}$. The new trajectory encoder $f_\tau(\cdot|\theta_\tau)$ only consists of a fully-connected network and a GRU. The policy of each subtask $\phi_i$ is a fully-connected network $f_{\phi_i}(\cdot; \theta_{\phi_i})$ with parameters $\theta_{\phi_i}$. Then, for each agent $g_a \in A_{\phi_i}$ that solves the subtask $\phi_i$, we feed its action-observation history $h_{\tau_a}$ into the subtask policy $f_{\phi_i}(\cdot; \theta_{\phi_i})$ and generate the individual Q-value $Q_a$. To associate the policy of each subtask with its representation, we utilize a subtask decoder $f_d(\cdot|\theta_d)$ conditioning on the subtask representation to generate the policy parameters of each subtask. The subtask decoder $f_d(\cdot|\theta_d)$ with parameters $\theta_d$ is just a single-layer fully-connected network, which maps $x_{\phi_i}$ to $\theta_{\phi_i}$ for each subtask $\phi_i$. By virtue of the differences between subtask representations, such subtask decoder can further increase the diversity of subtask policies.

In our implementation, for each agent $g_a$, we feed $h_{\tau_a}$ into all subtask policies to get the individual Q-value $Q_{a,\phi_i}$ for every subtask $\phi_i$. The individual Q-value $Q_a$ is the sum of $[Q_{a,\phi_i}]_{i=1}^k$ weighted by the one-hot identity of the selected subtask. If agent $g_a$ selects subtask $\phi_j$, $Q_a$ is actually equal to $Q_{a,\phi_j}$. Hence, each agent only trains the policy parameters of its selected subtask. In this way, agents with similar abilities tend to select the same subtask and thus can share their experiences to accelerate training and improve performance. Moreover, on the tasks that require to assign different subtasks to agents with similar trajectories, since each agent's observation input contains its one-hot identity, LDSA could learn to put more weights on the agent's identity input, and make the agent's identity

input become the main factor of the output subtask selection distribution. Then two agents can have distinct subtask selection distributions even if they have similar observation from the environment.

### 3.4 Overall training and inference

Similar to value function factorization methods [4–8], we use a mixing network with parameters $\theta_w$ to map all agents' individual Q-values $(Q_a, \boldsymbol{Q}_{-a})$ into the global Q-value $Q_{tot}$, where $Q_a$ is the individual Q-value of agent $g_a$ and $\boldsymbol{Q}_{-a}$ are the individual Q-values of other agents. In this work, we adopt the mixing network introduced by QMIX [5] thanks to its simple structure and superior performance. It can be easily extended to other more complex mixing networks [7, 8]. Then, all the parameters $\theta := (\theta_e, \theta_h, \theta_\tau, \theta_d, \theta_w)$ of our framework can be optimized by minimizing the TD loss of $Q_{tot}$ as follows:

$$\mathcal{L}_{\text{TD}}(\theta) = \mathbb{E}_{\mathcal{D}} \left[ \left( r + \gamma \max_{\mathbf{u}'} Q_{tot}(\boldsymbol{\tau}', \mathbf{u}'; \theta^-) - Q_{tot}(\boldsymbol{\tau}, \mathbf{u}; \theta) \right)^2 \right], \tag{7}$$

where $\theta^-$ are the parameters of the target network that are periodically copied from $\theta$. Considering the two regularizers in Eq. 4 and Eq. 6, the overall optimization objective of LDSA is:

$$\mathcal{L}(\theta) = \mathcal{L}_{\text{TD}}(\theta) + \lambda_\phi \mathcal{L}_\phi(\theta_e) + \lambda_h \mathcal{L}_h(\theta_e, \theta_h), \tag{8}$$

where $\lambda_\phi$ and $\lambda_h$ are positive coefficients of the two regularizers, respectively. During the inference phase (*i.e.*, test the decentralized policy), each agent $g_a$ selects the subtask with the maximum probability on the subtask selection distribution, *i.e.*, $\arg\max_{\phi_i} p(\phi_i | x_{\tau_a}, x_\Phi)$ and then chooses a greedy action according to the individual Q-value $Q_a$ for decentralized execution.

## 4 Experiments

In this section, we conduct several experiments to investigate the following questions: (1) Can LDSA improve the performance compared to the baselines? (Sec. 4.2) (2) How do the two proposed regularizers influence the performance? (Sec. 4.3) (3) Whether the superior performance of our method comes from the increase in the number of parameters? If not, which component contributes the most to our method? (Sec. 4.3) (4) Can LDSA learn dynamic subtask assignment and group agents reasonably? If so, agents with similar abilities solve the same subtask and each subtask has specific responsibility. (Sec. 4.4)

### 4.1 Experimental setup

**Environment** We evaluate LDSA on the SMAC benchmark [27], a challenging benchmark for cooperative MARL. There are two armies of units in the SMAC environment. Each ally unit is controlled by a decentralized agent that can only act based on its local observation and the enemy units are controlled by built-in handcrafted heuristic rules. The goal of the MARL algorithm is to maximize the test win rate for each battle scenario. In this paper, we adopt the default environment settings for SMAC. The version used in this work is SC2.4.10. We consider all 14 scenarios on the SMAC benchmark that can be classified into three different levels of difficulties: *Easy*, *Hard* and *Super Hard* scenarios.

**Baselines** We select QMIX [5], ROMA [33] and RODE [26] as our baselines. All of the baselines and LDSA belong to value function factorization methods and use the QMIX-style mixing network [5] for a fair comparison. Therefore, LDSA differs from the baselines only by the individual Q-value networks. QMIX is a natural baseline in which all agents share the individual Q-value network. ROMA and RODE are two recent works that learn implicit and explicit task decomposition in MARL, respectively. We implement all the baselines using their open-source codes based on PyMARL [27]. Note that, our experiments are conducted under the assumption that there is no prior knowledge about the environment. As mentioned in Sec. 1, RODE may not work when some basic actions are necessary for all subtasks. In the SMAC environment, there are four actions representing the agents' movement in four cardinal directions, which are important for all subtasks. To be effective,

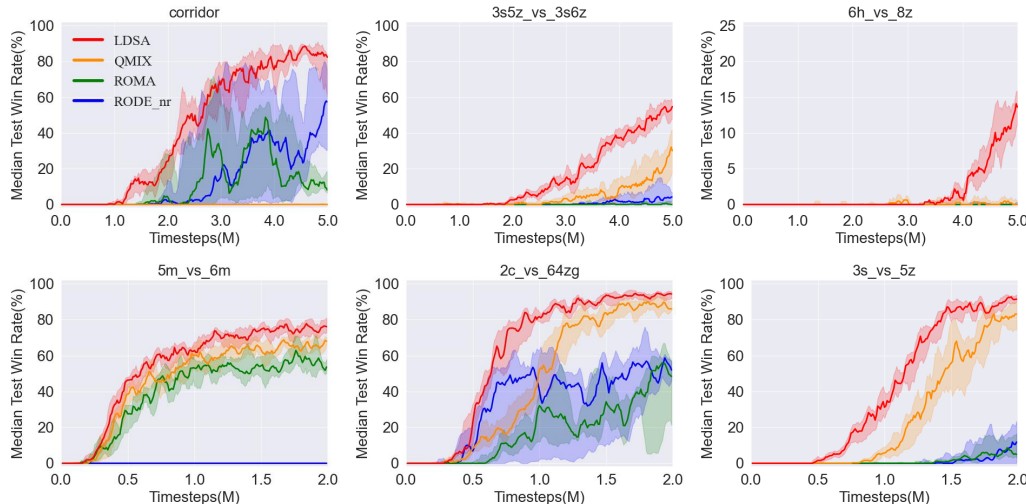

Figure 2: Comparison of our method against baselines on three *Super Hard* SMAC scenarios: `corridor`, `3s5z_vs_3s6z`, `6h_vs_8z` and three *Hard* SMAC scenarios: `5m_vs_6m`, `2c_vs_64zg`, `3s_vs_5z`. The solid line shows the median test win rate across 5 seeds and the shaded areas correspond to the 25-75% percentiles.

RODE uses several rules based on prior knowledge of the environment in its implementation, which manually enforce the four actions of moving be available to some subtasks forever under certain conditions. Therefore, to not involve prior knowledge, we remove these rules when we implement RODE. We use "RODE_nr" to represent RODE without manual rules in our experiments.

**Hyperparameters** For all experiments, the number of subtasks is set to 4 and the length of subtask representations is set to 64, *i.e.*, $k = 4$, $m = 64$. We carry out a grid search for regularizers' coefficients $\lambda_\phi$ and $\lambda_h$ on the SMAC scenario `corridor` and then set them to $10^{-3}$ and $10^{-3}$, respectively, for all scenarios. All the common hyperparameters of our method and baselines are set to be the same as that in the default implementation of PyMARL. We use lightweight network structures for the subtask encoder, subtask decoder, trajectory encoder and subtask policy. The detailed hyperparameters and network structures will be provided in Appendix A.

## 4.2 Performance on SMAC

We compare LDSA with the baselines across all 14 scenarios on the SMAC benchmark [27]. For every scenario, we carry out 5 independent runs with 5 different random seeds for all methods and show the median performance and 25-75% percentiles. Fig. 3 presents the averaged median test win rate across all 14 scenarios. We can observe that LDSA significantly improves the learning performance and surpasses about 7% median test win rate averaged across all 14 scenarios.

Fig. 2 shows the learning curves of LDSA and the baselines on six *Hard* or *Super Hard* scenarios. The results on the other eight scenarios are shown in Appendix B. LDSA outperforms all the baselines on all *Super Hard* and *Hard* scenarios with faster convergence. Specifically, on the scenarios that require diverse micromanagement techniques: `corridor`, `3s5z_vs_3s6z` and `6h_vs_8z`, the test win rate of LDSA exceeds that of the baselines at least 15%. On the *Easy* scenarios, LDSA shows similar performance as QMIX, which indicates that our method may not improve the learning performance on simple tasks not requiring task decomposition. Moreover, ROMA that learns implicit task decomposition performs even worse than QMIX on most of the scenarios. Without using rules based on prior knowledge, RODE fails to learn efficient

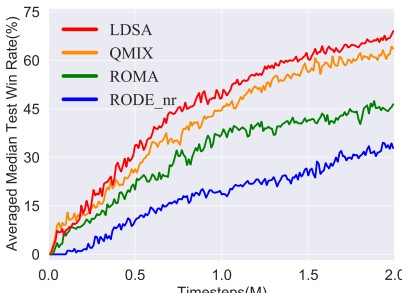

Figure 3: The averaged median test win rate across all 14 scenarios on the SMAC benchmark.

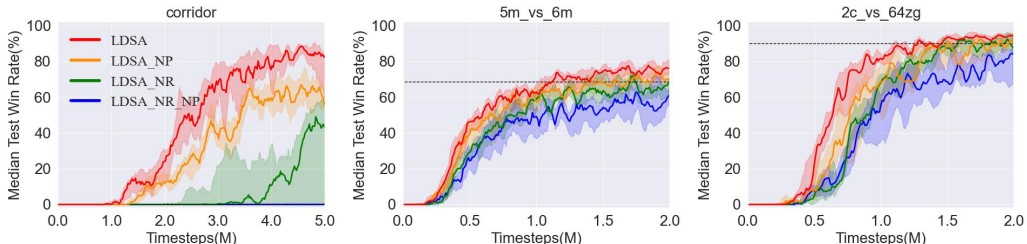

Figure 4: Ablation studies regarding the two proposed regularizers. "NP" and "NR" mean to remove $\mathcal{L}_h$ and $\mathcal{L}_\phi$ from the overall optimization objective of LDSA, respectively. The best performance of QMIX is shown as a dashed line.

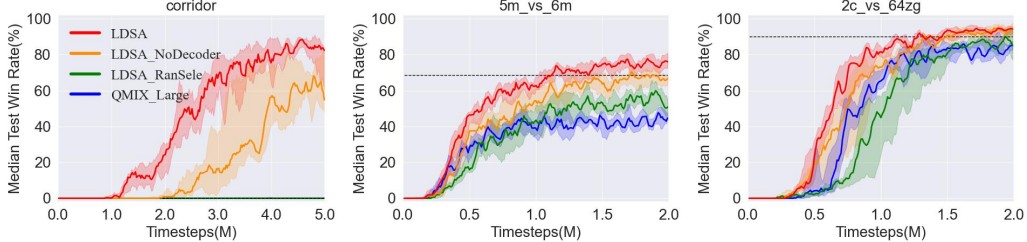

Figure 5: Ablation studies regarding components of LDSA. "LDSA_NoDecoder" represents LDSA without the subtask decoder. "LDSA_RanSele" indicates that LDSA randomly selects a subtask for each agent at each timestep. "QMIX_Large" means to increase the number of parameters in QMIX to be similar as that in LDSA. The best performance of QMIX is shown as a dashed line.

policies for subtasks, which demonstrates that discovering subtasks based on joint action space decomposition and restricting the action space of subtasks may severely limit the learning of subtask policies. These results further confirm that LDSA can learn effective task decomposition with dynamic subtask assignment to solve complex tasks. In Appendix D, we compare LDSA with another baseline CDS [16] that celebrates diversity among agents, and show the benefits of LDSA to balance the training complexity and the diversity of agent behavior. We also evaluate LDSA on Google Research Football (GRF) [20] to demonstrate the effectiveness of LDSA on various multi-agent tasks in Appendix E.

### 4.3 Ablation studies

In this subsection, we conduct ablation studies on three scenarios: `corridor`, `5m_vs_6m` and `2c_vs_64zg`, to show the effect of the two regularizers and test which component contributes most to LDSA. Fig. 4 reports the performance of LDSA when we remove each or both of the two regularizers $\mathcal{L}_h$ and $\mathcal{L}_\phi$. It can be observed that the performance of LDSA without both of the two regularizers is even worse than that of QMIX. Either of the two regularizers can improve performance and $\mathcal{L}_\phi$ improves more than $\mathcal{L}_h$, which indicates that keeping differences between subtasks is more important while $\mathcal{L}_h$ can stabilize training.

To figure out why LDSA performs better than QMIX, we investigate the contributions of three components of LDSA: the increase in the number of parameters, the ability-based subtask selection and the subtask decoder. As shown in Fig. 5, QMIX with similar parameters as LDSA can't improve performance and even may make training slower and degrade performance, which proves that the superior performance of LDSA is not due to the increase in the number of parameters. The performance of LDSA without the subtask decoder is lower than that of LDSA, which demonstrates that associating subtask policy with subtask representation benefits the performance. What's more, the large margin between the performance of LDSA with random subtask selection and that of LDSA reveals that the ability-based subtask selection contributes the most to our method. In summary, the superior performance of LDSA is largely due to the efficient subtask assignment. We also study the effect of the number of subtasks in Appendix C and conduct two more ablations for LDSA in Appendix F.

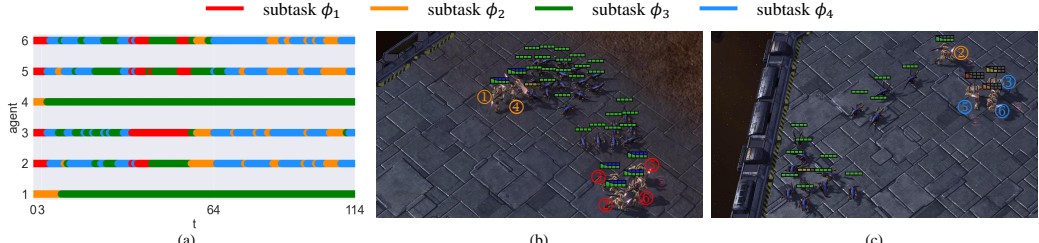

Figure 6: Visualizations of dynamic subtask assignment in one episode (114 timesteps) on `corridor`. Different colors indicate different assigned subtasks. (a) The subtask assignment for six ally agents along the whole episode. (b) and (c) show the game screenshots at $t = 3$ and $t = 64$, respectively, where each ally agent is marked by a colored number. The number represents the agent identity and the color indicates the assigned subtask.

### 4.4 Visualization of dynamic subtask assignment

In this subsection, we visualize the dynamic subtask assignment in one episode on the SMAC scenario `corridor` as shown in Fig. 6. The scenario `corridor` consists of 6 ally Zealots and 24 enemy Zerglings. Due to the huge disparity in the number of allies and enemies, it's hard to win if all ally Zealots rush to attack the enemy. Therefore, at the beginning ($t = 3$), our method learns to sacrifice two allies (agents $g_1$ and $g_4$) to attract most enemies and then the other 4 allies (agents $g_2$, $g_3$, $g_5$ and $g_6$) can focus fire to kill a small part of enemies. After that, the remaining enemy Zerglings controlled by heuristic rules will go through a narrow corridor, and stay at the bottom left corner of the scenario if there are no Zealots within the sight range. In the middle of the episode ($t = 64$), the remaining enemy forces are still strong, our method assigns the healthiest ally (agent $g_2$) to draw out a small number of enemies and kill them with the other allies (agents $g_3$, $g_5$ and $g_6$). This process will repeat until all enemies are killed. Moreover, we find that our method will assign the dead allies (agents $g_1$ and $g_4$) to a fixed subtask (subtask $\phi_3$), which can prevent them from interfering with the learning of other subtasks. Besides, it is worth noting that subtask $\phi_1$ and subtask $\phi_4$ have similar responsibilities that agents focus fire to kill enemies, but under different situations related to the agent's health point. When an agent needs to focus fire to kill enemies, it tends to select subtask $\phi_1$ if it has a good health point otherwise select subtask $\phi_4$. These visualizations demonstrate the effectiveness and rationality of the dynamic subtask assignment in our method.

## 5 Related work

**Value-based MARL** Value-based MARL algorithms have achieved great progress in recent years. IQL [34] simply trains an independent Q-value network for each agent, which treats the other agents as part of the environment. This method may not converge due to the non-stationarity of the environment caused by the changing policies of other agents. The other extreme method is to treat all agents as a single agent and learn a global Q-value based on the joint action-observation space, which alleviates the non-stationarity but suffers from the scalability challenge, as the joint action-observation space grows exponentially with the number of agents. To trade off these two methods, most of the value-based MARL algorithms factorize the global Q-value into individual Q-values for centralized training and decentralized execution. VDN [4] and QMIX [5] factorize the global Q-value by additivity and monotonicity, respectively. QTRAN [6] transforms the original global Q-value function into an easily factorizable one to expand the representation capacity. QPLEX [8] proposes a duplex dueling network architecture to implement the complete IGM [6] function class. These methods focus on designing the mixing network. There are other value-based works studying MARL from the perspective of communication [35, 36], exploration [37, 38] and robustness [39, 40].

**Parameter sharing in MARL** To learn efficiency and scalability, most of the MARL works employ the technique of parameter sharing among agents [4–15]. Terry et al. [17] demonstrate that parameter sharing can effectively alleviate the non-stationarity problem in MARL. Despite its efficiency, fully-shared parameters among agents may destroy the diversity of agent behavior in many complex multi-agent tasks [20, 27]. To trade off experience sharing and behavioral diversity among agents, SePS [21] groups agents based on their identities during pretraining and each group

shares one policy for training. CDS [16] proposes to decompose the policy of each agent as the sum of a shared part and a non-shared part. In this work, we don't fix the group each agent can share with. Each agent in our method can adaptively choose a group to share based on its trajectory during training, which enables dynamic parameter sharing.

**Task decomposition in MARL**     Task decomposition plays an important role in many complex real-world multi-agent systems, such as software engineering [41], healthcare [42] and traffic management [43]. Once the multi-agent task is decomposed, agents can be assigned to the restricted subtasks that are easier to solve, and thus the learning complexity is greatly reduced. However, it is challenging to come up with a set of subtasks that can effectively decompose the whole multi-agent task. The most straightforward way is to predefine the subtasks by leveraging the prior domain knowledge [22–25]. But the prior knowledge may not be available in many uncertain environments. To solve this problem, ROMA [33] introduces the concept of roles for each agent based on its local observation and conditions agents' policies on their roles. Nguyen et al. [44] redesign the first layer of the mixing network in QMIX [5] and view the output nodes of the first layer as different roles. These two works focus on implicit task decomposition. RODE [26] explicitly defines the roles/subtasks based on joint action space decomposition during pretraining, where each subtask corresponds to a subset of actions. In this work, we learn vector representations for subtasks that can automatically adapt to the environment during training.

## 6    Conclusion

Task decomposition is an important approach to simplify complex multi-agent tasks and has not been well solved without using prior knowledge. To this end, we propose to decompose the task into several subtasks represented by latent embeddings. Agents select subtasks according to their abilities, which are indicated by their action-observation histories. In this way, agents dealing with the same subtask can share their learning to solve the subtask, which can learn the specific abilities required by all subtasks under a tractable training complexity. Although the embedding representation for each subtask may be abstract, it essentially clusters agents with similar abilities into the same group and thus agents can dynamically share their experiences to accelerate training and improve performance. After training, each agent is aware of all subtask policies and can adaptively choose the most appropriate subtask policy to execute based on its ability. The empirical results further demonstrate that LDSA learns effective task decomposition with dynamic subtask assignment and significantly improves the learning performance compared to the baselines. We hope our method could provide a new perspective on task decomposition and subtask learning in MARL.

## Acknowledgments

This work was supported by the National Natural Science Foundation of China under Contract 61836011 and 62021001, and by the Huawei Cloud project "Multi-agent Competitive Decision Scenario Algorithm and Technology Research". It was also supported by GPU cluster built by MCC Lab of Information Science and Technology Institution, USTC.

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
