# OpenReview forum: "LDSA: Learning Dynamic Subtask Assignment in Cooperative Multi-Agent Reinforcement Learning"
_NeurIPS.cc/2022/Conference — NeurIPS 2022 Accept_

### Official Review · Reviewer_i6UD · 2022-07-09

**Rating:** 6
**Confidence:** 4
**Soundness:** 3 good
**Presentation:** 4 excellent
**Contribution:** 2 fair

**Summary:**

The paper proposes an algorithm to learn dynamic subtask assignment (LDSA) in cooperative MARL. First, it uses a subtask encoder to construct a vector representation for each subtask. Then, it proposes an ability-based subtask selection strategy to dynamically group agents into the similar subtask. In addition, the authors employ two tricks to make the representations of subtasks distinguishable and the learning process steady. Empirical results show that LDSA learns reasonable and effective subtask assignment for better collaboration and significantly improves the learning performance on the challenging StarCraft II micromanagement benchmark.

**Questions:**

- We wonder how this method would perform if the multi-agent cooperative tasks do not require diverse agents' policies. For example, in Level-Based Foraging (GitHub - semitable/lb-foraging: Level-based Foraging (LBF): A multi-agent environment for RL), agents need to cooperate in strong coordination to collect foods with different levels where different types of policies are not necessary. We wonder if the effectiveness of LDSA depends on the type of problem.
- We wonder why the performances of ROMA and RODE are considerably worse than those in their original paper. For example, RODE can achieve above 80% winning rate on the map of corridor and 6h\_vs\_8z, while in this paper, RODE cannot learn anything after 5 million timesteps.
- Are the trajectory encoders shown in Figure 1(b) & 1(c) the same?
- As the authors claim in the main text, the agents learn different abilities after training, so we wonder how these agents could perform tasks different from the training phase, which are known as ad-hoc tasks.


**Limitations:**

The authors do not claim the limitations of their method in the paper. We think, as mentioned above that the effectiveness of LDSA depends on the type of problem it aims to solve since the diversity of policies is not directly related to the effectiveness of cooperation. The authors should carry out more experiments on other different environments.

**Strengths And Weaknesses:**

Strengths:
- The writing of the paper is clear and structured.
- Previous works dealing with multi-agent cooperative tasks predefine the task decomposition using prior domain knowledge, while this work does not use prior knowledge to decompose the task.
- The number of subtasks should be set in advance but is a robust hyperparameter that all the scenarios in the SMAC benchmark can share an identical number of subtasks.
- The ablation study demonstrates the effectiveness of the two regularizations and different components of LDSA.

Weaknesses:
- Since the number of subtasks $k$ can vary, the subtask defined in Definition 1 is more self-defined than objectively existing, making the definition informal. We think the authors should clarify the assumption under which the overall task can be decomposed into subtasks.
- Ground truth about the subtasks is not available, so we prefer that this method works on the SMAC benchmark because it encourages the diversity of agents’ policies which encourages exploration. We think it would be better if the authors could clarify the assumption that the more diverse the policies, the better the agents would perform.
- The experiments are only carried out on the SMAC benchmark, which restricts the reliability of the results. As far as we know, after fine-tuning, even a baseline like QMIX can achieve excellent performance on SMAC (GitHub - hijkzzz/pymarl2: Fine-tuned MARL algorithms on SMAC (100% win rates on most scenarios)). We believe it would make the results more convincing if the authors could conduct more experiments on other types of environments.

---

> ### Author Response · Authors · 2022-08-02
> **Response to Reviewer i6UD - Part 2**
>
> **Q3**: We wonder how this method would perform if the multi-agent cooperative tasks do not require diverse agents' policies. For example, in Level-Based Foraging (GitHub - semitable/lb-foraging: Level-based Foraging (LBF): A multi-agent environment for RL), agents need to cooperate in strong coordination to collect foods with different levels where different types of policies are not necessary. We wonder if the effectiveness of LDSA depends on the type of problem.
>
> **A3**: In lines 265~268,  we mentioned that LDSA shows similar performance as QMIX on simple tasks not requiring task decomposition. QMIX can be approximated as a special case of LDSA when all agents learn to only select the same one subtask. So LDSA will perform similarly as QMIX on the multi-agent cooperative tasks that do not require diverse agents' policies.
>
> **Q4**: We wonder why the performances of ROMA and RODE are considerably worse than those in their original paper. For example, RODE can achieve above 80% winning rate on the map of corridor and 6h\_vs\_8z, while in this paper, RODE cannot learn anything after 5 million timesteps.
>
> **A4**: For ROMA, it used the old version of SMAC benchmark in its original paper, while we use the latest version. Performance is not always comparable between versions. Maybe ROMA is sensitive to hyper-parameters. So it performs not so good as that in the original paper when moving to a new version. For RODE, first, as we mentioned in lines 238~243, RODE uses several rules based on prior knowledge in its implementation to help decompose the task. To avoid involving prior knowledge, we remove these rules when we implement RODE. Second, we follow the default hyper-parameter setting of PyMARL for LDSA and all baselines, while RODE in the original paper doesn't. For example,  we use $\epsilon$-greedy for exploration. The $\epsilon$ anneals linearly from 1.0 to 0.05 over 50k timesteps and keeps constant for the rest of the learning for all SMAC maps in our paper, while RODE in the original paper extends the epsilon annealing time from 50k to 500k on three hard exploration maps 3s5z_vs_3s6z, 6h_vs_8z, and 27m_vs_30m.
>
> **Q5**: Are the trajectory encoders shown in Figure 1(b) & 1(c) the same?
>
> **A5**:  The trajectory encoders shown in Figure 1(b) & 1(c) are different. We will distinguish the two trajectory encoders more clearly in the next revision.
>
> **Q6**: As the authors claim in the main text, the agents learn different abilities after training, so we wonder how these agents could perform tasks different from the training phase, which are known as ad-hoc tasks.
>
> **A6**: In our method, agents can learn different abilities that are required to solve the training task. If different tasks have one or two same subtasks, perhaps we can transfer the same subtask ability from the training task to a different testing task. But we do not intend to transfer our method between different tasks because two different tasks usually have different sizes of state-action space. We will explore how to transfer our method between different tasks in the future.
>
> Finally, thank you again for your thoughtful comments. We will incorporate your suggestions into our next revision.  If some of your concerns are addressed, you could consider raising the rating. This is very important for us and we will appreciate it very much.
>
> **Reference**
>
> - [1] Kurach K, Raichuk A, Stańczyk P, et al. Google research football: A novel reinforcement learning environment[C], Proceedings of the AAAI Conference on Artificial Intelligence. 2020, 34(04): 4501-4510.

---

> ### Author Response · Authors · 2022-08-02
> **Response to Reviewer i6UD - Part 1**
>
> We thank the reviewer for the detailed comments. We hope we can address your concerns below.
>
> **Q1**:  We think the authors should clarify the assumption under which the overall task can be decomposed into subtasks. We think it would be better if the authors could clarify the assumption that the more diverse the policies, the better the agents would perform.
>
> **A1**: Thanks for your suggestion. We will clarify these assumptions in the next revision.
>
> **Q2**:  The experiments are only carried out on the SMAC benchmark, which restricts the reliability of the results. As far as we know, after fine-tuning, even a baseline like QMIX can achieve excellent performance on SMAC (GitHub - hijkzzz/pymarl2: Fine-tuned MARL algorithms on SMAC (100% win rates on most scenarios)). We believe it would make the results more convincing if the authors could conduct more experiments on other types of environments.
>
> **A2**:  We conduct more experiments on two challenging Google Research Football (GRF) [1] academy scenarios  *academy_3_vs_1_with_keeper* and *academy_counterattack_easy*. We show the mean and standard deviation of the test score reward across five random seeds as follows.
>
> Table 1: Comparison of our method against baselines on the GRF scenario academy_3_vs_1_with_keeper.
>
> | Timesteps |      ROMA       |      RODE       |      QMIX       |      LDSA       |
> | :-------: | :-------------: | :-------------: | :-------------: | :-------------: |
> |  **1M**   | $0.04 \pm 0.03$ | $0.03 \pm 0.03$ | $0.03 \pm 0.03$ |  $0.1 \pm 0.1$  |
> |  **2M**   | $0.01 \pm 0.02$ | $0.1 \pm 0.08$  | $0.12 \pm 0.08$ | $0.57 \pm 0.21$ |
> |  **3M**   | $0.02 \pm 0.03$ | $0.1 \pm 0.14$  | $0.32 \pm 0.22$ | $0.71 \pm 0.12$ |
>
> Table 2: Comparison of our method against baselines on the GRF scenario academy_counterattack_easy.
>
> | Timesteps |      ROMA       |     RODE      |      QMIX       |      LDSA       |
> | :-------: | :-------------: | :-----------: | :-------------: | :-------------: |
> |  **1M**   |  $0.0 \pm 0.0$  | $0.0 \pm 0.0$ | $0.04 \pm 0.05$ | $0.05 \pm 0.05$ |
> |  **2M**   | $0.01 \pm 0.01$ | $0.0 \pm 0.0$ | $0.03 \pm 0.04$ | $0.11 \pm 0.14$ |
> |  **3M**   |  $0.0 \pm 0.0$  | $0.0 \pm 0.0$ | $0.14 \pm 0.18$ | $0.27 \pm 0.17$ |
> |  **4M**   |  $0.0 \pm 0.0$  | $0.0 \pm 0.0$ | $0.29 \pm 0.26$ | $0.48 \pm 0.08$ |
>
> We can observe that LDSA could also achieve better performance than baselines on GRF, which demonstrates the effectiveness of LDSA on various multi-agent tasks.

---

> > ### Comment · Reviewer_i6UD · 2022-08-05
> > **Thanks for your responses**
> >
> > Thanks for responding. Added experiments improve the reliability of the results. However, the assumptions I was concerned about are not well clarified. Moreover, the results of RODE are pretty different from other papers like [1, 2, 3], in which the authors also carry out experiments on SMAC 2.4.10. If the performance heavily relies on the hyperparameters, I believe the authors should also conduct experiments of LDSA with the hyperparameters aligned with the original RODE to eliminate the effects of hyperparameters. I keep the score.
> >
> > [1] Li J, Kuang K, Wang B, et al. Deconfounded Value Decomposition for Multi-Agent Reinforcement Learning. In ICML, pages 12843-12856, 2022.
> >
> > [2] Naderializadeh N, Hung F H, Soleyman S, et al. Graph convolutional value decomposition in multi-agent reinforcement learning. arXiv preprint arXiv:2010.04740, 2020.
> >
> > [3] Li P, Tang H, Yang T, et al. PMIC: Improving Multi-Agent Reinforcement Learning with Progressive Mutual Information Collaboration. In ICML, pages 12979-12997, 2022.

---

> > > ### Author Response · Authors · 2022-08-05
> > > **Why the results of RODE in our paper are poor?**
> > >
> > > We have run the original code of RODE before, we find that the performance of RODE heavily relies on the manual rules rather than the hyperparameters.  RODE uses these manual rules based on prior knowledge of the environment in its implementation to help decompose the task. For example, on the SMAC super hard maps corridor and 3s5z_vs_3s6z, RODE without manual rules performs much worse than RODE with manual rules (i.e, original RODE). We show the median test win rate (%) across five random seeds as follows.
> > >
> > > corridor:
> > >
> > > | Timesteps | RODE with rules (original RODE) | RODE without rules (RODE in our paper) |
> > > | :-------: | :-----------------------------: | :------------------------------------: |
> > > | **2.5M**  |              29.9               |                  2.4                   |
> > > | **5.0M**  |              81.5               |                  57.3                  |
> > >
> > > 3s5z_vs_3s6z:
> > >
> > > | Timesteps | RODE with rules (original RODE) | RODE without rules (RODE in our paper) |
> > > | :-------: | :-----------------------------: | :------------------------------------: |
> > > | **2.5M**  |              3.5              |                  0.0                  |
> > > | **5.0M**  |              80.2              |               4.2   |
> > >
> > > [1, 2, 3] don't focus on task decomposition like RODE and our method. They just treat RODE as a general framework and extend their methods with RODE. Of course, they use the default manual rules in RODE. Our method is to learn task decomposition without using prior knowledge and our experiment is conducted under the assumption that there is no prior knowledge. So we remove the manual rules in the original RODE. This is the main reason why the results of RODE in our paper are pretty different from other papers. We will replace the "RODE" in the experimental section with "RODE w/o rules" for better understanding in the next version.

---

> > > > ### Comment · Reviewer_i6UD · 2022-08-07
> > > > **More questions about definition and assumptions**
> > > >
> > > > Thanks for the supplementary experiments on RODE. However, I still hope the authors could make a more formal clarification on the definition of sub-tasks and claim the assumptions under which their algorithms could take effect in a more comprehensive way.

---

> > > > > ### Author Response · Authors · 2022-08-09
> > > > > **Response about the definition and assumptions**
> > > > >
> > > > > Thank you again for replying to our rebuttal. We have uploaded a rebuttal revision according to your suggestions.
> > > > >
> > > > > **Q1**：Since the number of subtasks k can vary, the subtask defined in Definition 1 is more self-defined than objectively existing, making the definition informal. We think the authors should clarify the assumption under which the overall task can be decomposed into subtasks.
> > > > >
> > > > > **A1**：Thank you for suggestion. We have given a more formal definition of sub-tasks in the revision as follows.
> > > > >
> > > > > **Definition 1** (Subtasks). For a fully cooperative multi-agent task $G = \langle A, S, U, P, r, Z, O, \gamma \rangle$, we assume there exists a set of $k$ subtasks, denoted as $\Phi \equiv \{\phi_1, \phi_2, \cdots, \phi_k\}$, where $k \in \mathbb{N}^+$ is unknown and we consider $k$ as a tunable hyperparameter. Each subtask $\phi_i$ is defined by a tuple $\langle x_{\phi_i}, G_{\phi_i}, \pi_{\phi_i} \rangle$, where $i \in \{1, 2, \cdots, k\}$ is the identity of subtask, $x_{\phi_i} \in \mathbb{R}^m$ is a vector representation (or latent embeddings) of subtask $\phi_i$, $G_{\phi_i} = \langle A_{\phi_i}, S, U, P, r, Z, O, \gamma \rangle$ is the Dec-POMDP model of subtask $\phi_i$, and $\pi_{\phi_i} : T \times U \rightarrow [0, 1]$ is the policy of subtask $\phi_i$. $A_{\phi_i}$ is the set of agents assigned to subtask $\phi_i$ and each agent can only select one subtask to solve at each timestep, i.e., $\bigcup_{i=1}^k A_{\phi_i} = A$ and $A_{\phi_i} \bigcap A_{\phi_j} = \varnothing$ if $i \neq j$. Each agent $g_a \in A_{\phi_i}$ shares the policy parameters of $\pi_{\phi_i}$.
> > > > >
> > > > > **Q2**：Ground truth about the subtasks is not available, so we prefer that this method works on the SMAC benchmark because it encourages the diversity of agents’ policies which encourages exploration. We think it would be better if the authors could clarify the assumption that the more diverse the policies, the better the agents would perform.
> > > > >
> > > > > **A2**: Under the condition that there is no prior domain knowledge, our method aims to learn potential task decomposition in MARL. To this end, we propose to learn latent embeddings for subtasks.  It's hard for us to determine whether the learned subtask representations are correct due to the lack of ground truth about the subtasks. So we visualize the dynamic subtask assignment on the SMAC scenario corridor and find that agents can indeed learn different subtasks, which demonstrates the effectiveness and rationality of our method. We agree that the most direct reason why our method works is that it encourages the diversity of agents’ policies which encourages exploration. But we don't think that the more diverse the policies, the better the agents would perform. It's a better way to trade off the diversity and sharing among agents.
> > > > >
> > > > > Finally, we sincerely thank you for taking the time to reply to us. Your suggestions make the paper more clear and solid. Hopefully, our rebuttal and new revision could properly address your concerns, and we will appreciate it if you could upgrade your score.

---

> > > > > > ### Comment · Reviewer_i6UD · 2022-08-09
> > > > > > **Thanks for your responses.**
> > > > > >
> > > > > > I appreciate authors for addressing my concerns and will raise my score from 5 to 6.

---

> > > > > > > ### Author Response · Authors · 2022-08-09
> > > > > > > **Thank you for raising the score**
> > > > > > >
> > > > > > > We are happy that we could address you concerns, and we feel very grateful that you can raise the score.

---

### Official Review · Reviewer_z7kN · 2022-07-10

**Rating:** 6
**Confidence:** 4
**Soundness:** 3 good
**Presentation:** 3 good
**Contribution:** 2 fair

**Summary:**

This work addresses skill/ability specialisation for agents sharing a controller in cooperative MARL. LDSA, the proposed approach in this work, assumes agents' skill specialisation is required to solve subtasks constituting a particular task. LDSA subsequently learns to assign agents to subtasks by:
1) Creating identically sized representations of subtasks and agents' abilities.
2) Match agents and subtasks according to the similarity of their representations.
3) Learning a policy that agents need to follow for each subtask.

The process of learning representations of agents & subtasks, alongside the optimal policy for each subtask, is done end-to-end in combination with QMIX, which is a value-based value factorisation MARL method. Aside from the temporal difference loss functions associated with QMIX, LDSA's optimised function also includes regularisation terms that (a) force subtasks to have different representations from each other and (ii) agents to have similar subtask assignment distributions between consecutive timesteps.

This work subsequently evaluates LDSA in various scenarios in SMAC. LDSA's performance is compared against QMIX and previous MARL methods for role discovery, such as ROMA and RODE. This work also conducted an ablation study to find the importance of the QMIX-based objective function alongside other regularisation terms for the trained agents' returns. These experiments' results empirically demonstrate LDSA's significantly improved performance compared to previous role assignment methods. Further analysis of the resulting agents' behaviour in the SMAC corridor scenario indicates agents' ability to solve diverse tasks to perform well in the environment.

**Questions:**

**Questions**

1. How does LDSA deal with the need for assigning different roles to agents with similar observed trajectories?
2. How will LDSA perform compared to a method that directly estimates the mixture weights of each subtask policy as opposed to computing it based on agent ability-subtask representational similarity.
3. What happens if LDSA uses a mixture of subtask policies (that are weighted according to inferred component weights) as opposed to sampling one subtask policy at any timestep?
4. In Figure 6 on Section 4.4, what is the difference between the subtask denoted by the red and blue lines?
5. In the problem formulation, why does subtasks have separate transition function compared to the solved task? Isn't it the case that tasks and subtasks are situated in the same environment (and hence sharing the same transition function)?

**Minor suggestions for writing**

1. Consider using different names for the two trajectory encoders in Lines 160 and 185.
2. For each environment used in evaluation, consider mentioning the different roles that are required for solving each environment.

**Limitations:**

As indicated by the feedback provided in the previous sections, the authors have not highlighted the potential limitations of their work. Furthermore, I believe no statement regarding potential negative social impact of the work is required since this work is currently a generic MARL method that has yet been applied to real-world problems.

**Strengths And Weaknesses:**

**Originality**

**(Minor Weakness) LDSA is an incremental idea relative to the most recent works in the area.**

The idea of assigning agents to roles based on their observed trajectories has been explored before in the ROMA approach (Wang et al., 2020). While ROMA proposes an implicit role assignment method that conditions agents' policies on task representations, explicit approaches like LDSA that learn K separate policies for different subtasks and the optimal mixture of these policies has recently been explored by Nguyen et al. (2022). Compared to the idea from Nguyen et al. (2022), LDSA merely offers a different way of computing the weights and utilisation of policies of each subtask.

**Quality**

**(Major Strength) Comparison against several prior methods in role assignments show LDSA's significantly higher returns.**

By outperforming ROMA and RODE, this work demonstrates that LDSA has the potential of improving several existing approaches for role assignment.

**(Major Strength) Empirical evaluation of learned subtasks.**

Empirical evaluation of learned subtasks in SMAC's corridor setup indicates that LDSA manages to learn meaningful subtask assignment in some environments. This experiment further illustrates the significance of this work's results and should ideally be done for all evaluated environments.

**(Major Weakness) Missing baselines on skill/ability specialisation for parameter sharing in MARL.**

Since the abstract and introduction section frames LDSA as a solution for skill/ability specialisation under parameter sharing in MARL, a comparison against CDS (Li et al., 2021) is necessary since it also attempts to achieve the same goal as LDSA.

Beyond the similarity in their underlying goals, comparing LDSA against CDS may also help investigate the effects of one of LDSA's potential weaknesses. In particular, LDSA assigns a role to agents based on their observed trajectories. This can be a drawback when agents exhibiting similar trajectories must be assigned different subtasks to improve overall performance. For instance, consider a kitchen setup where two agents must clean dirty plates, chop different types of ingredients, and serve the chopped ingredients on the cleaned plates. Under this setup, the optimal policy is for both agents to jointly wash the dirty dishes, divide the chopping of different ingredients among them, and subsequently serve the ingredient they have chopped on a plate. Note that before ingredients are chopped, the trajectory of both agents is similar since they are solving the same task up to that point. This lack of a symmetry-breaking mechanism to assign agents with similar trajectories to different subtasks is not present in CDS since each agent in CDS has individual components not shared with others.

**(Major Weakness) Missing baseline to highlight the importance of the proposed role assignment method.**

Since LDSA main novelty is its proposed role assignment method, the baselines can be useful to highlight the importance of LDSA's role/subtask assignment components:

1. A baseline where task assignment probability is directly estimated from agents' observed trajectories. Comparing LDSA's performance against this baseline helps us understand the importance of mapping subtasks & agent trajectories to fixed-length representations and computing their similarities to compute task assignment probability.
2. A baseline where an agent employs a mixture of the individual subtasks' policies rather than using a single sampled policy.

Note that these two aspects are the main differences between LDSA and the approach proposed by Nguyen et al. (2022). Thus, comparing LDSA's performance against these may provide important insights regarding which role assignment approach is optimal.

**(Minor Weakness) Lack of diversity in environments used for evaluation.**

While this only provides a minor improvement, the generality of this work's results can be further demonstrated by showing it works in a wider range of environments. For instance, Google Research Football and Overcooked are two environments where subtask assignment is required for learning.

**Clarity:**

**(Major Strength) The work is generally well written and very easy to understand.**

Other minor suggestions to improve clarity will be provided in the questions/suggestions section below.

**Significance:**

**(Major Strength) The results of this work has potential significance to people focusing on dynamic task assignment in MARL.**

The idea of jointly learning subtask and ability representations can be an essential contribution if it is demonstrated to perform better compared to the baselines mentioned above. However, further experiments are required to demonstrate the significance of this work.

**Citations:**

Wang et al. 2020. ROMA: Multi-Agent Reinforcement Learning with Emergent Roles. ICML 2020

Wang et al. 2021. RODE: learning roles to decompose multi−agent tasks. ICLR 2021.

Nguyen et al. 2022. Learning to Transfer Role Assignment Across Team Sizes. AAMAS 2022.

Li et al. 2021. Celebrating Diversity in Shared Multi-Agent Reinforcement Learning. NeurIPS 2021.

---

> ### Author Response · Authors · 2022-08-02
> **Response to Reviewer z7kN - Part 2**
>
> **Q4**: How does LDSA deal with the need for assigning different roles to agents with similar observed trajectories?
>
> **A4**: In the training phase, each agent samples a subtask from the subtask selection distribution for training. So it's possible for two agents to sample two different subtasks even if the two agents have the exact same subtask selection distribution. In other words, LDSA can explore assigning different roles to agents with similar observed trajectories during training. In the testing phase, each agent selects the subtask with the maximum probability on the subtask selection distribution. Since each agent's observation input contains its one-hot identity, if all agents have similar observed trajectories and we need to assign different roles to them, LDSA could learn to put more weights on the agent's identity input and then the agent's identity input could become the main factor of the output subtask selection distribution. In this way, two agents can have distinct subtask selection distributions even if they have similar observation from the environment.
>
> **Q5**: How will LDSA perform compared to a method that directly estimates the mixture weights of each subtask policy as opposed to computing it based on agent ability-subtask representational similarity. What happens if LDSA uses a mixture of subtask policies (that are weighted according to inferred component weights) as opposed to sampling one subtask policy at any timestep?
>
> **A5**: Thanks for your suggestion. We conduct two more ablations for LDSA, named LDSA_DireProb and LDSA_Mix, respectively. LDSA_DireProb means to directly estimate the subtask selection distribution from agents' observed trajectories. LDSA_Mix represents that each agent uses a mixture of subtask policies as opposed to sampling one subtask policy. We compare these two ablations with LDSA on the SMAC super hard map *corridor*.  We show the mean and standard deviation of the test win rate (%) across five random seeds as follows.
>
> Table 3: Ablation studies regarding components of LDSA.
>
> | timesteps | LDSA_DireProb |    LDSA_Mix     |      LDSA       |
> | :-------: | :-----------: | :-------------: | :-------------: |
> | **1.5M**  | $0.0 \pm 0.0$ | $6.6 \pm 10.4$  | $11.4 \pm 7.8$  |
> | **3.0M**  | $0.0 \pm 0.0$ | $31.3 \pm 31.4$ | $58.9 \pm 23.6$ |
> | **4.5M**  | $0.0 \pm 0.0$ | $48.2 \pm 27.8$ | $85.4 \pm 7.0$  |
>
> We can see that the performance of LDSA_Mix is lower than that of LDSA, which indicates that using a single subtask policy for each agent is better than the mixture of subtask policies. In other words, it's better for each agent to learn one subtask at each timestep rather than a mixture of all subtasks. Besides, the comparison between LDSA and LDSA_DireProb highlights the importance of computing the subtask selection distribution based on similarity of agent trajectories and subtask representations.
>
> **Q6**: In Figure 6 on Section 4.4, what is the difference between the subtask denoted by the red and blue lines?
>
> **A6**: In Figure 6, subtask $\phi_1$ (red) and  subtask $\phi_4$ (blue) have similar responsibilities that agents focus fire to kill a small part of enemies, but under different situations. Subtask $\phi_1$ appears at the beginning of the episode when agent's health point is good, while subtask $\phi_4$ mainly appears in the middle and late of the episode when agent has a low health point. In other words, when an agent needs to focus fire to kill enemies, it tends to select subtask $\phi_1$ if it has a near full heal point otherwise select subtask $\phi_4$.
>
> **Q7**: In the problem formulation, why does subtasks have separate transition function compared to the solved task? Isn't it the case that tasks and subtasks are situated in the same environment (and hence sharing the same transition function)?
>
> **A7**: In MARL, a task is usually specialized by its transition function and reward function. Therefore, to distinguish different subtasks, we assume that different subtasks may have distinct transition functions and reward functions. And the definition of subtask transition function doesn't affect our method because our method doesn't need to use the transition function of subtask.  It's an abstract definition to some extent.
>
> Finally, thank you again for your thoughtful comments. We will incorporate your suggestions into our next revision. If some of your concerns are addressed, you could consider raising the rating. This is very important for us and we will appreciate it very much.
>
> **Citations:**
>
> Wang et al. 2020. ROMA: Multi-Agent Reinforcement Learning with Emergent Roles. ICML 2020
>
> Wang et al. 2021. RODE: learning roles to decompose multi−agent tasks. ICLR 2021.
>
> Nguyen et al. 2022. Learning to Transfer Role Assignment Across Team Sizes. AAMAS 2022.
>
> Li et al. 2021. Celebrating Diversity in Shared Multi-Agent Reinforcement Learning. NeurIPS 2021.

---

> ### Author Response · Authors · 2022-08-02
> **Response to Reviewer z7kN - Part 1**
>
> We thank the reviewer for the detailed comments. We hope we can address your concerns below.
>
> **Q1**: LDSA learns K separate policies for different subtasks and the optimal mixture of these policies has recently been explored by Nguyen et al. (2022). Compared to the idea from Nguyen et al. (2022), LDSA merely offers a different way of computing the weights and utilisation of policies of each subtask.
>
> **A1**: There are great differences between Nguyen et al. (2022) and LDSA. First, Nguyen et al. (2022) focuses on designing the first layer of the mixing network and aims to transfer the mixing network across different team sizes, while LDSA focuses on designing the individual networks and aims to balance the training complexity and the diversity of agent behavior. Second, in Nguyen et al. (2022), each agent has its own policy and the policy of each role/subtask is a mixture of all agents' policies, while in LDSA,  each subtask has its own policy and each agent selects one subtask policy as its policy according to its ability. Third, Nguyen et al. (2022) has no explicit definition for role/subtask, while LDSA gives.
>
> **Q2**: Since the abstract and introduction section frames LDSA as a solution for skill/ability specialisation under parameter sharing in MARL, a comparison against CDS (Li et al., 2021) is necessary since it also attempts to achieve the same goal as LDSA.
>
> **A2**: LDSA focuses more on how to decompose tasks rather than how to share parameters.  We didn't choose CDS (Li et al., 2021) as a baseline because CDS (Li et al., 2021) is not a method that focuses on task decomposition. Here, we compare LDSA and other baselines with CDS (Li et al., 2021) on two challenging Google Research Football (GRF) academy scenarios  *academy_3_vs_1_with_keeper* and *academy_counterattack_easy*. We show the mean and standard deviation of the test score reward across five random seeds as follows.
>
> Table 1: Comparison of our method against baselines on the GRF scenario academy_3_vs_1_with_keeper.
>
> | Timesteps |      ROMA       |      RODE       |      QMIX       |       CDS       |      LDSA       |
> | :-------: | :-------------: | :-------------: | :-------------: | :-------------: | :-------------: |
> |  **1M**   | $0.04 \pm 0.03$ | $0.03 \pm 0.03$ | $0.03 \pm 0.03$ | $0.07 \pm 0.08$ |  $0.1 \pm 0.1$  |
> |  **2M**   | $0.01 \pm 0.02$ | $0.1 \pm 0.08$  | $0.12 \pm 0.08$ | $0.09 \pm 0.11$ | $0.57 \pm 0.21$ |
> |  **3M**   | $0.02 \pm 0.03$ | $0.1 \pm 0.14$  | $0.32 \pm 0.22$ | $0.35 \pm 0.26$ | $0.71 \pm 0.12$ |
>
> Table 2: Comparison of our method against baselines on the GRF scenario academy_counterattack_easy.
>
> | Timesteps |      ROMA       |     RODE      |      QMIX       |       CDS       |      LDSA       |
> | :-------: | :-------------: | :-----------: | :-------------: | :-------------: | :-------------: |
> |  **1M**   |  $0.0 \pm 0.0$  | $0.0 \pm 0.0$ | $0.04 \pm 0.05$ | $0.01 \pm 0.01$ | $0.05 \pm 0.05$ |
> |  **2M**   | $0.01 \pm 0.01$ | $0.0 \pm 0.0$ | $0.03 \pm 0.04$ | $0.01 \pm 0.02$ | $0.11 \pm 0.14$ |
> |  **3M**   |  $0.0 \pm 0.0$  | $0.0 \pm 0.0$ | $0.14 \pm 0.18$ | $0.08 \pm 0.08$ | $0.27 \pm 0.17$ |
> |  **4M**   |  $0.0 \pm 0.0$  | $0.0 \pm 0.0$ | $0.29 \pm 0.26$ | $0.2 \pm 0.18$  | $0.48 \pm 0.08$ |
>
> We can observe that LDSA outperforms CDS (Li et al., 2021) and other baselines on both GRF scenarios,  which demonstrates the effectiveness of LDSA on various multi-agent tasks.
>
> **Q3**: While this only provides a minor improvement, the generality of this work's results can be further demonstrated by showing it works in a wider range of environments. For instance, Google Research Football and Overcooked are two environments where subtask assignment is required for learning.
>
> **A3**: Thanks for your suggestion. We also validate the benefits of LDSA on the challenging Google Research Football (GRF), see A2. As for the overcooked, we find it is an interesting environment for fully cooperative human-AI task performance and we didn't know it before. We will try to evaluate our method on it in the future.

---

> > ### Comment · Reviewer_z7kN · 2022-08-07
> > **Response to Author Rebuttal (Part 1)**
> >
> > Thank you for replying to my questions/points in the original review. I appreciate the additional experiments that were done to address my previous concerns. While my rating currently remains the same, I am willing to improve it if the authors can convince me of the soundness of their experiments (both in the original paper and the ones recently included in their rebuttal) by replying to these follow-up points.
> >
> > **Important Points (potentially improves scores if addressed)**
> >
> > **1. (Related to A2) Inclusion of CDS as a baseline**
> >
> > While from a narrative standpoint CDS (Li et al., 2021) is indeed framed as an approach for finding a good balance between the benefits and drawbacks of parameter sharing, it is still a means towards behaviour specialisation to improve performance in cooperative MARL. Experiments conducted in the CDS paper demonstrate that agents' specialised behaviour emerging from training via CDS corresponds to them concurrently executing different subtasks/roles to perform well in the evaluated environments.
> >
> > Actionable item:
> >
> > Since it will not take much space, CDS should be included as a baseline in all experiments that are reported in the final paper.
> >
> > **2. (Related to A2 and A3) Scenario selection for Google Research Football (GRF)**
> >
> > I appreciate the addition of GRF results. However, I am unsure whether academy_3_vs_1_with_keeper or academy_counterattack_easy are the best scenarios for LDSA's experiments. Li et al. (2021) demonstrated that CDS does not significantly outperform QPLEX (which is not equipped with modifications to allow behaviour specification) in academy_3_vs_1_with_keeper. It is also unclear whether subtask decomposition is required to solve academy_counterattack_easy based on (i) its description in the GRF repository and (ii) LDSA and CDS not outperforming QMIX in the results.
> >
> > Actionable item:
> >
> > Choose one of these items and include it in the reply and final paper:
> > - Use GRF scenarios that more clearly highlights the need for task decomposition (e.g. academy_counterattack_hard as reported in the CDS paper) or,
> > - Explain why academy_counterattack_easy is a good scenario by highlighting the different subtasks that are required for solving this environment.
> >
> > **3. (Related to A2) LDSA outperforming CDS**
> >
> > Given the reported mean and standard deviations, I disagree about the point that LDSA outperforms CDS. CDS and LDSA's 95% confidence intervals still overlap. As the difference in performance between LDSA and CDS is not significant, I believe the authors must be careful of making such statements in the paper.
> >
> > **4. (Related to A4) Role assignment based on trajectory similarity**
> >
> > I see how symmetry-breaking in subtask assignment for agents with similar trajectories may occur with the introduction of one-hot agent id features. As simple as it is, I believe this is a solution to an important issue which should be highlighted more in the final paper. Note that the authors can make their argument shorter by omitting their point about sampling as a potential solution for symmetry-breaking during exploration. Between sampling and one-hot ids, I believe using one-hot ids is a better solution for symmetry-breaking.
> >
> > Actionable item:
> >
> > A4 should be included in the final paper.
> >
> > **5. (Related to A5) Ablations of LDSA**
> >
> > I appreciate the addition of LDSA's ablations. I believe these results should be added to the paper (even if it is in the appendix if the authors run out of space).
> >
> > Actionable item:
> > - (Clarification) Why is LDSA_DireProb not learning at all? It is even performing worse than ROMA, which also directly estimates role-related information (i.e. representation in the case of ROMA) via agent's trajectory using the role encoder.
> > - Adding comparison with ablations to main paper.

---

> > > ### Author Response · Authors · 2022-08-09
> > > **Response to the follow-up points**
> > >
> > > Thank you again for replying to our rebuttal. We have uploaded a rebuttal revision according to your suggestions.
> > >
> > > **Q1**: Since it will not take much space, CDS should be included as a baseline in all experiments that are reported in the final paper.
> > >
> > > **A1**: We have added the results on GRF in Appendix A.5. Due to time limit, we compare LDSA with CDS on there SMAC scenarios 5m_vs_6m, 10m_vs_11m, 27m_vs_30m in Appendix A.4. We observe that LDSA performs better than CDS on all three scenarios. Moreover, CDS can't learn anything on the scenario 27m_vs_30m that needs to control a large number of agents. This may be because CDS requires to learn one shared policy network and $n$ non-shared policy networks (i.e., a total of $n+1$ policy networks) if we have $n$ agents, which may lead to high training complexity when the multi-agent task contains a large number of agents. Our method only needs to learn $k$ subtask policies, where $k \ll n$ when controlling a large number of agents. Therefore, compared with CDS, LDSA can achieve a better balance between the training complexity and the diversity of agent behavior.
> > >
> > > **Q2**: Explain why academy_counterattack_easy is a good scenario by highlighting the different subtasks that are required for solving this environment.
> > >
> > > **A2**: In academy_counterattack_easy, we need to control 4 players against 1 defender and 1 goalkeeper. If all our 4 players rush forward to shoot, it's easy to cause offside or the ball is easily intercepted by the enemy goalkeeper.  To solve this environment, there are two main different subtasks. One is to dribble the ball to the top side of the field and attract the attention of the enemy players to create an enemy vacancy in the down side of the field. The other is to run to the down side of the field and wait for a long pass from teammates on the top side, and then shoot. We have added this in Appendix A.5.
> > >
> > > **Q3**: Given the reported mean and standard deviations, I disagree about the point that LDSA outperforms CDS. CDS and LDSA's 95% confidence intervals still overlap. As the difference in performance between LDSA and CDS is not significant, I believe the authors must be careful of making such statements in the paper.
> > >
> > > **A3**: Thanks for your reminder. We will be careful of making such statements.
> > >
> > > **Q4**: A4 should be included in the final paper.
> > >
> > > **A4**: Thank you for advice. We have added it in Sec. 3.3.
> > >
> > > **Q5**: Why is LDSA_DireProb not learning at all? It is even performing worse than ROMA, which also directly estimates role-related information (i.e. representation in the case of ROMA) via agent's trajectory using the role encoder. Adding comparison with ablations to main paper.
> > >
> > > **A5**: ROMA uses a role encoder based on agent's trajectory to generate the agent's policy parameters. LDSA_DireProb directly employs the agent's trajectory to decide which subtask/role to choose, but the policy parameters of each subtask/role still depend on the subtask/role representation rather than agent's trajectory (i.e., representation-dependent subtask policy). So if we have explicit subtask representation and the subtask policy depend on its representation, it's better for each agent to know all subtasks when choosing a subtask (i.e., select a subtask based on subtask representation). In other words, it's hard to learn to select the subtask if we have no information about the subtask. ROMA could do this because it has no explicit role/subtask representation and it only learns implicit roles. Besides, to enable efficient learning, ROMA proposes two regularizers to learn identifiable and specialized roles that only based on agent's trajectory. We have added these two more ablations in Appendix A.6.
> > >
> > > **Q6**: I appreciate the additional details regarding Figure 6. As long as this does not take too much space, it is good if the authors try to fit A6 into the main paper.
> > >
> > > **A6**: Thanks for your appreciation. We have added it in Sec. 4.4.
> > >
> > > Finally, we sincerely thank you for taking the time to reply to us. Your suggestions help us a lot and make our work better understanding and more solid. Hopefully, our rebuttal and new revision could properly address your concerns, and we will appreciate it if you could upgrade your score.

---

> > > > ### Comment · Reviewer_z7kN · 2022-08-09
> > > > **Updated Ratings**
> > > >
> > > > Thank you for addressing my concerns. I found the author's replies to have reasonably addressed my concerns. As such, I have decided to update my ratings for this paper.
> > > >
> > > > Also note that with the new addition of GRF results in the paper, it is worth mentioning this environment in the abstract even if the GRF results only appear in the appendix.

---

> > > > > ### Author Response · Authors · 2022-08-09
> > > > > **Thank you for updating the rating**
> > > > >
> > > > > We are happy that we could address you concerns. We feel very grateful that you can raise the score. Thanks for your reminder. We have mentioned the GRF in the abstract.

---

> > ### Comment · Reviewer_z7kN · 2022-08-07
> > **Response to Author Rebuttal (Part 2)**
> >
> > This comment addresses the remaining points that have not been covered in the previous one. **The points in this comment generally has little to no bearing to my final ratings.**
> >
> > **6. (Related to A1) Comparison against Nguyen et al. (2022)**
> >
> > While I understand the three arguments made by the authors in A1, I believe that the first argument actually shows that the approach from Nguyen et al. (2022) is potentially more general since it can be applied to scenarios that have teams with different sizes. For the remaining arguments that show the difference between LDSA and the approach from Nguyen et al. (2022), I still view these differences as minor differences which cannot significantly increase my ratings regarding the originality of this work.
> >
> > Nevertheless, this issue regarding originality is not a major concern as long as the authors compare LDSA with an approach using some ideas from Nguyen et al. (2022) (as they did in the ablation experiments done in A5).
> >
> > **7. (Related to A6) Difference between subtasks visualised in Figure 6**
> >
> > I appreciate the additional details regarding Figure 6. As long as this does not take too much space, it is good if the authors try to fit A6 into the main paper.
> >
> > **8. (Related to A7) Subtask formalisation**
> >
> > I understand that the way subtasks are formalised has no bearing towards the proposed method. Nonetheless, it remains uncanny to have the transition and reward function change between subtasks. The transition function of an environment (i.e. the way environment state changes as a result of agents' joint actions) and the reward function in most examples I can think of remains the same between subtasks.
> >
> > A more natural subtask formalisation can be formulated if the authors view it as options in the semi-Markov Decision Process (SMDP) framework (Sutton et al., 1999), which does not assume that the transition/reward function changes between options. The only difference with SMDPs is then how different agents may concurrently execute different subtasks/options since agents are working in a multiagent system.

---

### Official Review · Reviewer_VLj9 · 2022-07-25

**Rating:** 8
**Confidence:** 4
**Soundness:** 4 excellent
**Presentation:** 4 excellent
**Contribution:** 4 excellent

**Summary:**

This paper introduces an algorithm for efficiently training diverse abilities/policies in cooperative multi agent tasks. It learns to decompose a task into subtasks, match agents to a subtask based on their action-observation trajectory, and train policies for each subtask from the experience of agents assigned to it on that timestep. It does better than QMIX, ROMA and RODE on the harder tasks on the StarCraft II micromanagement tasks.

**Questions:**

- In the ablations section: Is there a way to decrease LDSA parameters and compare to the original-sized QMIX rather than increase QMIX parameters?


**Limitations:**

Limitations - they mentioned it may not improve things on simple tasks. I would also add that this is formulated only for a fully cooperative game at the moment.
The author didn't discuss potential negative social impacts, it would be good to mention even if the answer is inconclusive.

**Strengths And Weaknesses:**

Strengths:
- Well organized and clearly written paper.
- Original and elegant method for decomposing subtasks without prior knowledge, and also without limiting the action space per subtask (like RODE) which is more flexible.
- It does beat the other algorithms on harder tasks, which is a significant contribution.
- The ablations, and other investigations (of e.g. effect of number of subtasks) helpful for understanding the different parts of the algorithm.

Weaknesses:
- Not sure why comparing only to QMIX style mixing networks makes for a fair comparison; why would comparing to other types of approaches not be fair?

---

> ### Author Response · Authors · 2022-08-02
> **Response to Reviewer VLj9**
>
> We thank the reviewer for the detailed comments. We hope we can address your concerns below.
>
> **Q1**: Not sure why comparing only to QMIX style mixing networks makes for a fair comparison; why would comparing to other types of approaches not be fair?
>
> **A1**:  LDSA focuses on designing the individual network of each agent and can use any type of mixing network. Therefore, to demonstrate the effectiveness of LDSA, we use the same mixing network for all methods and then study the benefits of the individual networks of different methods. In our paper, we use the QMIX-style mixing network. It's also a fair comparison if we use any other mixing networks.
>
> **Q2**: In the ablations section: Is there a way to decrease LDSA parameters and compare to the original-sized QMIX rather than increase QMIX parameters?
>
> **A2**: Yes, there is. To decrease LDSA parameters to be similar as that in the original-sized QMIX, we can decrease the number of nodes in the hidden layer of two trajectory encoders in Figure 1(b) and 1(c), and reduce the length of subtask representations.
>
> **Q3**: The author didn't discuss potential negative social impacts, it would be good to mention even if the answer is inconclusive.
>
> **A3**:  Although we haven’t evaluated our method on real applications, we believe our work doesn't have potential negative societal impacts.
>
> Finally, thank you again for your recognition and positive review to our work. We will incorporate your suggestions into our next revision.

---

### Official Review · Reviewer_7hXy · 2022-07-26

**Rating:** 5
**Confidence:** 4
**Soundness:** 2 fair
**Presentation:** 3 good
**Contribution:** 3 good

**Summary:**

Sharing parameters among all agents usually limits the behavioral diversity while learning a separate policy for each agent without any parameter sharing usually leads to high training complexity.
To balance the training complexity and the diversity of agents’ behaviors, this paper proposes a dynamic parameter sharing framework LDSA. LDSA divides the agents into up to $k$ groups (where $k$ is a manually set parameter) according to their action-observation histories. All agents share the parameters of the input layer and the GRU layer of the individual Q network. For the output layer, only the agents within the same group share the same parameters and the agents in different groups use different parameters. Experimental results in SMAC valid the benefits of the proposed dynamic parameter sharing mechanism.

**Questions:**

Please see the weaknesses above.

**Limitations:**

The group number $k$ has to be manually set aforehand for each task.

**Strengths And Weaknesses:**

**Strengths**
  * The paper is well motivated and clearly written.
  * The proposed framework LDSA is technically sound.

**Weaknesses**
  * (1) The proposed method has little to do with "dynamic task assignment". The core of LDSA is dividing the agents into up to $k$ groups based on their different behaviors and making the agents in different groups do not share parameters. There are no explicitly pre-defined tasks for the agents to solve. Therefore, using the words "dynamic parameter sharing according to agents' behaviors" may be more appropriate.
  * (2) Important related works and baselines are missing. [1] and [2] propose different methods to balance the training complexity and the diversity of agents’ behaviors by dynamically sharing parameters among agents. I notice that the authors cited these two works in the paper but not discussed and compared with them. These two works should be well discussed in the related work and compared in the experiments.
  * (3) The experimental evaluations in SMAC is not convincing. Recently, [3] and [4] have verified that the optimized QMIX (completely sharing the parameters among all agents) could achieve 100% win rates on all Easy, Hard and Super Hard scenarios of SMAC. Therefore, SMAC may not be a good testbed to validate the benefits of the dynamic parameter sharing mechanism. The authors could conduct more experiments in more complex environments, e.g., the challenging Google Research Football (GRF) [5].


**Reference**
  * [1] Christianos F, Papoudakis G, Rahman M A, et al. Scaling multi-agent reinforcement learning with selective parameter sharing[C],
  * International Conference on Machine Learning. PMLR, 2021: 1989-1998.
  * [2] Chenghao L, Wang T, Wu C, et al. Celebrating diversity in shared multi-agent reinforcement learning[J]. Advances in Neural Information Processing Systems, 2021, 34: 3991-4002.
  * [3] Hu J, Jiang S, Harding S A, et al. Rethinking the implementation tricks and monotonicity constraint in cooperative multi-agent reinforcement learning[J]. arXiv e-prints, 2021: arXiv: 2102.03479.
  * [4] Hao X, Wang W, Mao H, et al. API: Boosting Multi-Agent Reinforcement Learning via Agent-Permutation-Invariant Networks[J]. arXiv preprint arXiv:2203.05285, 2022.
  * [5] Kurach K, Raichuk A, Stańczyk P, et al. Google research football: A novel reinforcement learning environment[C], Proceedings of the AAAI Conference on Artificial Intelligence. 2020, 34(04): 4501-4510.

---

> ### Author Response · Authors · 2022-08-02
> **Response to Reviewer 7hXy - Part 2**
>
> **Q3**: The experimental evaluations in SMAC is not convincing. Recently, [3] and [4] have verified that the optimized QMIX (completely sharing the parameters among all agents) could achieve 100% win rates on all Easy, Hard and Super Hard scenarios of SMAC. Therefore, SMAC may not be a good testbed to validate the benefits of the dynamic parameter sharing mechanism. The authors could conduct more experiments in more complex environments, e.g., the challenging Google Research Football (GRF) [5].
>
> **A3**: In recent years, SMAC may be the most commonly used multi-agent benchmark. [3] and [4] could achieve 100% win rates on all Easy, Hard and Super Hard scenarios of SMAC because both of them implement their code based on [PyMARL2](https://github.com/hijkzzz/pymarl2) (the fine-tuned PyMARL), which adds lots of tricks such as TD($\lambda$), large batch size, death agent masking and so on. A simple algorithm can also obtain great performance if equipped with lots of tricks.  And we think adding tricks  could be approximated as reducing the tasks' difficulty. Therefore, we implement LDSA and the baselines based on the official [PyMARL](https://github.com/oxwhirl/pymarl), which doesn't use these tricks. All the common settings of LDSA and baselines use the default settings in PyMARL. We believe that this can also demonstrate the effectiveness of our method. We also validate the benefits of LDSA on the challenging Google Research Football (GRF), see A2.
>
> Finally, thank you again for your thoughtful comments. We will incorporate your suggestions into our next revision.  If some of your concerns are addressed, you could consider raising the rating. This is very important for us and we will appreciate it very much.
>
> **Reference**
>
> - [1] Christianos F, Papoudakis G, Rahman M A, et al. Scaling multi-agent reinforcement learning with selective parameter sharing[C], International Conference on Machine Learning. PMLR, 2021: 1989-1998.
> - [2] Chenghao L, Wang T, Wu C, et al. Celebrating diversity in shared multi-agent reinforcement learning[J]. Advances in Neural Information Processing Systems, 2021, 34: 3991-4002.
> - [3] Hu J, Jiang S, Harding S A, et al. Rethinking the implementation tricks and monotonicity constraint in cooperative multi-agent reinforcement learning[J]. arXiv e-prints, 2021: arXiv: 2102.03479.
> - [4] Hao X, Wang W, Mao H, et al. API: Boosting Multi-Agent Reinforcement Learning via Agent-Permutation-Invariant Networks[J]. arXiv preprint arXiv:2203.05285, 2022.
> - [5] Kurach K, Raichuk A, Stańczyk P, et al. Google research football: A novel reinforcement learning environment[C], Proceedings of the AAAI Conference on Artificial Intelligence. 2020, 34(04): 4501-4510.

---

> ### Author Response · Authors · 2022-08-02
> **Response to Reviewer 7hXy - Part 1**
>
> We thank the reviewer for the detailed comments. We hope we can address your concerns below.
>
> **Q1**: The proposed method has little to do with "dynamic task assignment". The core of LDSA is dividing the agents into up to k groups based on their different behaviors and making the agents in different groups do not share parameters. There are no explicitly pre-defined tasks for the agents to solve. Therefore, using the words "dynamic parameter sharing according to agents' behaviors" may be more appropriate.
>
> **A1**:  We think our method is not merely dynamic parameter sharing. Our motivation considers the fact that many complex multi-agent tasks consist of a set of subtasks. For example, a task decomposition (or subtask division) in football game could be forward, center, defender and goalkeeper. A good task decomposition can effectively reduce task complexity and allow better team cooperation. However, it's difficult to pre-define subtasks in a complex multi-agent task without using prior domain knowledge. To this end, we propose to learn abstract vector representations for subtasks, which is the core idea of LDSA.  To learn reasonable and effective subtasks,  we introduce the ability-based subtask selection strategy, representation-dependent subtask policy and two regularizers. After training, each agent is aware of all subtask policies and can adaptively choose the most appropriate subtask policy to execute based on its trajectory histories. Therefore, we believe our method could provide a new perspective for task decomposition and subtask learning, more than dynamic parameter sharing.
>
> **Q2**: Important related works and baselines are missing. [1] and [2] propose different methods to balance the training complexity and the diversity of agents’ behaviors by dynamically sharing parameters among agents. I notice that the authors cited these two works in the paper but not discussed and compared with them. These two works should be well discussed in the related work and compared in the experiments.
>
> **A2**: First, [1] groups agents based on their identities during pre-training and each group uses one policy. [2] decomposes the policy of each agent as the sum of a shared part and a non-shared part. In both [1] and [2], policy parameters that an agent should optimize are fixed throughout the training. On the contrary, each agent in our method could optimize different policy parameters of different subtasks during training. Therefore, we don't think [1] and [2] are methods of dynamically sharing parameters like our method. We will add a more detailed discussion about [1] and [2] in the next revision.  Second, we didn't choose [1] and [2] as baselines because [1] and [2] are not methods that focus on task decomposition. Specifically, [1] is a policy-based algorithm where each agent can receive an individual reward, while our method is a value-based algorithm where each agent can only receive a shared team reward. Here, we compare LDSA and other baselines with CDS [2] on two challenging Google Research Football (GRF) academy scenarios *academy_3_vs_1_with_keeper* and *academy_counterattack_easy*. We show the mean and standard deviation of the test score reward across five random seeds as follows.
>
> Table 1: Comparison of our method against baselines on the GRF scenario academy_3_vs_1_with_keeper.
>
> | Timesteps |      ROMA       |      RODE       |      QMIX       |       CDS       |      LDSA       |
> | :-------: | :-------------: | :-------------: | :-------------: | :-------------: | :-------------: |
> |  **1M**   | $0.04 \pm 0.03$ | $0.03 \pm 0.03$ | $0.03 \pm 0.03$ | $0.07 \pm 0.08$ |  $0.1 \pm 0.1$  |
> |  **2M**   | $0.01 \pm 0.02$ | $0.1 \pm 0.08$  | $0.12 \pm 0.08$ | $0.09 \pm 0.11$ | $0.57 \pm 0.21$ |
> |  **3M**   | $0.02 \pm 0.03$ | $0.1 \pm 0.14$  | $0.32 \pm 0.22$ | $0.35 \pm 0.26$ | $0.71 \pm 0.12$ |
>
> Table 2: Comparison of our method against baselines on the GRF scenario academy_counterattack_easy.
>
> | Timesteps |      ROMA       |     RODE      |      QMIX       |       CDS       |      LDSA       |
> | :-------: | :-------------: | :-----------: | :-------------: | :-------------: | :-------------: |
> |  **1M**   |  $0.0 \pm 0.0$  | $0.0 \pm 0.0$ | $0.04 \pm 0.05$ | $0.01 \pm 0.01$ | $0.05 \pm 0.05$ |
> |  **2M**   | $0.01 \pm 0.01$ | $0.0 \pm 0.0$ | $0.03 \pm 0.04$ | $0.01 \pm 0.02$ | $0.11 \pm 0.14$ |
> |  **3M**   |  $0.0 \pm 0.0$  | $0.0 \pm 0.0$ | $0.14 \pm 0.18$ | $0.08 \pm 0.08$ | $0.27 \pm 0.17$ |
> |  **4M**   |  $0.0 \pm 0.0$  | $0.0 \pm 0.0$ | $0.29 \pm 0.26$ | $0.2 \pm 0.18$  | $0.48 \pm 0.08$ |
>
> We can observe that LDSA outperforms CDS [2] and other baselines on both GRF scenarios, which demonstrates the effectiveness of LDSA on various multi-agent tasks.

---

> > ### Comment · Reviewer_7hXy · 2022-08-09
> > **Thank you for covering my questions**
> >
> > Hi authors,
> > thank you for the additional experimental evaluations on GRF and for addressing some of the points that I mentioned. I raise my score to 4 now.
> >
> > However, I agree with reviewer i6UD and I still have concerns about the experiments on SMAC. The fact is that, after setting proper parameters,  QMIX (with completely shared parameters among all agents) can achieve excellent performance on SMAC  (GitHub - hijkzzz/pymarl2: Fine-tuned MARL algorithms on SMAC (100% win rates on most scenarios)).
> >
> > The reviewer's point is that we cannot say a method is better than the compared baselines when setting relatively poor parameters for the baseline methods. Besides, recently, the official designers of SMAC have adopted pymarl2 as default implementations [1]. Therefore, in a revised version, I recommend the authors could re-evaluate LDSA based on pymarl2.
> >
> >
> > [1] SMACv2: A New Benchmark for Cooperative Multi-Agent Reinforcement Learning

---

> > > ### Author Response · Authors · 2022-08-09
> > > **Thanks for you reply**
> > >
> > > We sincerely thank you for taking the time to reply to us and raising the score. We agree that SMAC may be not an appropriate benchmark for MARL because the fine-tuned QMIX on pymarl2 can achieve 100% win rates on most scenarios. And it's difficult for other methods to further greatly improve performance based on pymarl2. As MARL researchers, we also struggled to find a new general MARL benchmark. But we find that many new multi-agent environments either are too simple or have no uniform standards. For example, the official Google Research Football (GRF) environment is not very suitable for the CTDE paradigm. In the default setting of GRF, each agent can obtain a global state of the environment, while in CTDE, each agent can only get a local observation. So we need to self-define the local observation for each agent and the global state for mixing network, which will vary in different papers. Therefore, so far, SMAC may be the most suitable benchmark for its uniform standards to CTDE paradigm. This is the reason why we choose the SMAC benchmark.  Thank you for introducing SMACv2 to us. We find that SMACv2 is a new recent benchmark that introduces more challenging scenarios compared to SMAC. We will try to evaluate our ideas on SMACv2 in the future.

---

> > > ### Author Response · Authors · 2022-08-09
> > > **About setting relatively poor parameters for the baseline methods**
> > >
> > > Hi,  reviewer 7hXy. The purpose of our work is not to achieve state-of-the-art performance on SMAC by fine-tuning lots of hyper-parameters like fine-tuned QMIX on PyMARL2. This involves a considerable amount of work. So we choose to demonstrate the effectiveness of our method by setting all the common hyper-parameters of our method and baselines as that in the default implementation of PyMARL. And we don't fine-tune the unique hyper-parameters of our method too much, such as the number of subtasks $k$. All the unique hyper-parameters of our method are set same for all scenarios.

---

> > > > ### Comment · Reviewer_7hXy · 2022-08-09
> > > > **Thanks for your responses.**
> > > >
> > > > Thank the authors for the rebuttal and the new revision. I upgrade the score to 5 now.

---

> > > > > ### Author Response · Authors · 2022-08-09
> > > > > **Thank you for improving the score**
> > > > >
> > > > > Thank you for replying to us again, and we feel very grateful that you can improve the score.

---

### Author Response · Authors · 2022-08-09
**Thanks for all Reviewers**

Thank all reviewers for your time and valuable suggestions. We submit a new revision and add your suggestions in it, where the added contents are marked in red. We hope our rebuttal and new revision could address your concerns. We would appreciate it if you could re-evaluate our submission and we are looking forward to discussions if you have any other concerns.

---

### Meta-Review · Area_Chair_LDUk · 2022-08-25

**Recommendation:** Accept
**Confidence:** Certain

**Metareview:**

This paper proposes a method to dynamically group agents with similar representations and assign subtasks to each group so that they can effectively share parameters among agents within the same group while specializing across groups. The results on StarCraft Micromanagement benchmark and Google Research Football domain show that the proposed method outperforms relevant baselines including QMIX, ROMA, and RODE.

The reviewers found that the idea is interesting and technically sound, and the paper is very well-written. Although there were several concerns about the lack of baselines (CDC) and the lack of challenging benchmarks, the authors addressed most of them during the rebuttal period by updating the results with additional baselines, an additional benchmark (Google Research Football), and additional ablation studies. As a result, all of the reviewers agreed that the result is significant enough to be presented at NeurIPS. Thus, I recommend accepting this paper.

**Award:**

No

---

### Decision · Program_Chairs · 2022-09-14

Accept